# Adherence to lipid-lowering guidelines for secondary prevention and potential reduction in CVD events in Swedish primary care: a cross-sectional study

Helena Ödesjö [1], Staffan Björck,[2] Stefan Franzén,[2] Per Hjerpe,[1] Karin Manhem,[3] Annika Rosengren,[3] Jörgen Thorn,[1] Samuel Adamsson Eryd[3]

**To cite:** Ödesjö H, Björck S, Franzén S, *et al.* Adherence to lipid-lowering guidelines for secondary prevention and potential reduction in CVD events in Swedish primary care: a cross-sectional study. *BMJ Open* 2020;**10**:e036920. doi:10.1136/bmjopen-2020-036920

¹Primary Health Care, School of Public Health and Community Medicine, Institute of Medicine, Sahlgrenska Academy, University of Gothenburg, Gothenburg, Sweden
²Centre of Registers Västra Götaland, Gothenburg, Sweden
³Department of Molecular and Clinical Medicine, Institute of Medicine, Sahlgrenska Academy, University of Gothenburg, Gothenburg, Sweden

**Correspondence to**
Dr Helena Ödesjö;
helena.odesjo@vgregion.se

## ABSTRACT

**Objectives** The protective effect of lipid-lowering treatment for secondary prevention after coronary heart disease (CHD) has been well documented. Current guidelines recommend a target level for low-density lipoprotein cholesterol (LDL-C) of ≤1.8 mmol/L. The aim was to describe lipid-lowering treatment patterns and to provide an estimate of the potential reductions in cardiovascular disease (CVD) events with improved adherence to guidelines.

**Design** Cross-sectional.

**Setting** Primary care in a large Swedish region.

**Participants** 37 120 patients with CHD in a Swedish regional primary care quality register (QregPV), by 31 December 2015.

**Primary and secondary outcome measures** Proportion of patients on statin treatment and proportion of patients achieving LDL-C ≤1.8 mmol/L. Estimated number of CVD events calculated for (1) current treatment, (2) improved treatment and (3) lowered LDL-C, based on applying rate reductions from meta-analyses of randomised trials to the potentially undertreated population. Risk estimation modelling was based on 52 042 patients in the same register on January 2011 followed for 5 years.

**Results** Of 37 120 patients, 18% reached LDL-C ≤1.8 mmol/L and 32% were not on statin treatment. Based on individual risks, the estimated number of CVD events in the study group over 5 years was 9209/37 120. If all patients without a statin or with less potent statin treatment were given atorvastatin 80 mg, an estimated reduction of CVD events by 14% (7901 vs 9209) was seen. If all patients achieved LDL-C ≤1.8 mmol/L, the number of events was estimated to be reduced by 18% (7577 vs 9209).

**Conclusion** One-third of patients with CHD in primary care were not on lipid-lowering treatment. Based on the assumption that included patients would react to statin therapy the same way as the patients in randomised trials, improved adherence to treatment guidelines could lead to a substantial reduction in new CVD events.

## Strengths and limitations of this study

► The study was based on all patients with coronary heart disease in primary care in a Swedish region.
► Data on comorbidities, filled prescriptions and outcomes were available for all patients.
► Risk model calculations were based on the same regional population as the predictions.
► Whether patients in our study react to statin therapy similarly as in the randomised trials in the meta-analysis used is not known.
► There were missing data regarding body mass index and smoking, known risk factors for cardiovascular disease.

cardiovascular disease (CVD) events irrespective of age.[1 2] Current evidence indicates an almost linear relationship between LDL-C level and risk of CVD.[3 4] Although CVD mortality has decreased by over 60% over the past few decades in Sweden, it still accounts for one out of every three deaths.[5]

Low adherence to statin treatment is associated with a greater risk of death and new CVD events.[6 7] The first years after a cardiovascular event, adherence to secondary preventive treatment is still high but diminishes over time.[8 9] Adverse events are rare but nevertheless pose barriers for statin treatment both in patients and physicians.[10–12]

Current European and Swedish guidelines recommend a treatment goal for LDL-C below 1.8 mmol/L in patients with established CHD.[13 14] There has been a gradual improvement in the proportion of patients in Sweden reaching the LDL-C target 1 year after an acute myocardial infarction (AMI), and recent data suggests a figure of 65% in 2017.[15] Nevertheless, less is known about long-term adherence to current guidelines among patients with CHD in primary care or how an improvement in that regard affects morbidity.

## INTRODUCTION

Lowering of low-density lipoprotein cholesterol (LDL-C) by statin treatment in patients with coronary heart disease (CHD) effectively reduces the risk of recurrent

The aim was to describe lipid-lowering treatment patterns and to provide an estimate of the potential reductions in CVD events with improved adherence to guidelines.

## METHODS

We performed an observational cross-sectional register study of all primary care patients with CHD in a large Swedish region as of 31 December 2015.

### Study basis

Västra Götaland (VGR) in south-west Sweden is a mixed urban and rural region with 1.7 million inhabitants (17% of the Swedish population). Since 2009, all individuals in the region have been enrolled at a primary care centre. The region has about 200 centres that are publicly funded, whether publicly or privately operated. Primary care centres report individual data for all patients with CHD to a regional primary care quality register (QregPV).

### Databases

This study proceeded from information obtained by linking data from QregPV, a regional administrative healthcare database (Vega), the National Patient Register (NPR), the Prescribed Drug Register and the Cause of Death Register.

QregPV contains information about all primary care patients, irrespective of age or severity of disease, with a diagnosis of hypertension and/or CHD in VGR, including blood pressure, lipid levels, smoking, height and weight recorded during appointments and reported on a monthly basis. Starting in 2009, reporting to the register has been mandatory for all primary care units in VGR.

Vega covers all primary care in VGR since 2000. Data include diagnoses, dates and professional categories. NPR covers all hospital discharge diagnoses in Sweden since 1987, as well as outpatient appointments in specialised care since 2001. This study included NPR data since 1997, when the International Classification of Disease 10th revision (ICD-10) was adopted. The Prescribed Drug Register contains information about all prescriptions filled since 1 July 2005. The Cause of Death Register consists of information about underlying and contributing causes of death since 1961.

All patients in QregPV with a diagnosis of CHD entered from 1 January 2010 to 31 December 2015 were included. Diagnoses were defined according to ICD-10: CHD I20-I25, AMI I21, stroke I61, I63 - I64, type 2 diabetes mellitus E10-14, heart failure I50 and atrial fibrillation and flutter I48. CVD was defined as AMI or stroke.

### Patients

Proceeding from the patient file, we created two cohorts:
► Study cohort (2015)—All patients with CHD in QregPV on 31 December 2015 (index date). Previous comorbidity until 31 December 2015 and prescriptions filled up to 120 days before the index date were collected from NPR, Vega and the Prescribed Drug Register.

► Risk estimation cohort (2011)—52 042 patients with CHD in QregPV on 1 January 2011 (index date). Previous comorbidity until the index date and new events corresponding to diagnoses of AMI and stroke until 31 December 2015 were added from NPR and Vega. Information about deaths was retrieved from the Cause of Death Register.

For both cohorts, medical data were collected in accordance with the last observation carried forward (LOCF) method. We used LOCF in order to include as many patients as possible since we considered using the last measured value a better approach then imputation or further exclusion of patients due to missing values. The latest entry of each variable was used up to 900 days before the index date.

### Risk estimation

The risk estimation cohort was used to create a model for the risk of recurrent CVD and all-cause mortality based on individual data for age, sex, diabetes mellitus, history of heart failure and/or atrial fibrillation, stroke or AMI during the previous year and treatment with acetylic salicylic acid. Statin treatment was not included since it is one of the factors we assess. Smoking and body mass index (BMI) were not included due to a large number of missing values. Blood pressure was not included due to missing data in 2010 and the J-curved association described in observational studies.[16]

### Prediction

We calculated the individual risk of CVD and death from all causes, for patients in the study cohort, using the risk prediction model. We performed two separate adjusted predictions and evaluated the effect of:
► Lowering LDL-C to 1.8 mmol/L for all patients with a higher level.
► Adding atorvastatin 40/80 mg to patients with no or less potent statin therapy and LDL-C >1.8 mmol/L.

For risk reduction associated with lowering of LDL-C, data from Cholesterol Treatment Trialists' (CTT) collaboration were used, assuming that our patients react to statin therapy the same way as the patients included in those randomised trials. Overall risk reduction for any major vascular event was 22% (rate ratio 0.78; 95% CI 0.76 to 0.80) per mmol/L reduction in LDL-C based on almost 170 000 individuals.[3] Risk reduction values for separate outcomes in the CTT study differed. Risk reduction per mmol/L LDL-C reduction for all-cause mortality was 0.91 based on results from another CTT collaboration study.[17] Risk reduction with intensified statin treatment proceeded from expected percentage LDL-C lowering related to type of statin and dosage based on studies concerning statin effect on LDL-C level.[18 19]

### Statistical methods

Descriptive statistics are presented as arithmetic mean and SD for continuous variables and with frequencies and percentages for categorical variables. A Cox regression model was applied to the risk estimation cohort. The variables included in the model were age, sex, diabetes mellitus,

history of heart failure, history of atrial fibrillation, treatment with acetyl salicylic acid and stroke or AMI during the previous year. Age was modelled by means of a smoothing spline function with 4 df since the effect of age was not assumed to be linear. Sensitivity analyses with different models including BMI, smoking and systolic blood pressure were performed in order to test for robustness.

Time to event was right censored at death as end of follow-up for total mortality and time to CVD. The time at risk for each patient was defined as the time before the first event over a period of 5 years.

The survival function and risk of having experienced an event over the past 5 years were calculated. Summing up individual risks yielded the total number of individuals predicted to have experienced an event over the past 5 years. Adjustments were made such that the cumulative hazard was recalculated based on an expected reduced risk for events related to lowering of LDL-C directly or expected lowering as a result of more intense statin treatment. The cumulative hazard of an event was reduced by 22% per mmol/L LDL-C lowering. This model did not take competing risks into account, assuming that all patients were alive after 5 years if they did not suffer an event.

The statistical analyses used R 3.4.0 and SAS V.9.4 (SAS Institute).

### Patient and public involvement

Patients or the public were not involved in the design, conduct, reporting or dissemination plans of our research.

### RESULTS

Descriptive statistics for the study cohort are shown in table 1, there was a total of 57 341 patients, 37 120 of whom had information about LDL-C at baseline. Fewer than 20% (6747/37 120) achieved the LDL-C target of ≤1.8 mmol/L. Among the non-controlled patients, with LDL-C >1.8 mmol/L, an average LDL-C reduction of 1.2 mmol/L (40%) would be required to reach the LDL-C target. The subset of patients with AMI or stroke in the previous year had a mean age of 74.2 (10.5); 78% received statin treatment and 25% reached the LDL-C target.

Statin treatment was most prevalent for age 50–75, approximately 70% of whom filled a prescription, see figure 1. Prescriptions filled for statins diminish rapidly after the age of 80.

LDL-C for uncontrolled patients was predicted to decrease from a mean of 3.0–2.2 mmol/L if all patients received atorvastatin 40 mg and to 2.0 mmol/L with atorvastatin 80 mg. For all patients in the study cohort the corresponding levels were 2.7 to 2.1 and 1.9 mmol/L, see LDL-C and predicted LDL-C distribution in figure 2.

Total predicted CVD events for 5 years in the study cohort was 9209 (24.8%), see figure 3. A decrease of LDL-C to the target (≤1.8 mmol/L) or intensified statin treatment (all patients with less efficient therapy receive atorvastatin 80 mg) was estimated to result in a reduction of expected CVD events for 5 years by 1632 and 1308. This corresponds to a decreased event risk of 18% and 14%. Including only

| Table 1 | Characteristics of patients in the study cohort | | |
|---|---|---|---|
| Variable | All patients (n=37 120) | LDL-C ≤1.8 mmol/L (n=6747) | LDL-C >1.8 mmol/L (n=30 373) |
| Age | 73.0 (10.1) | 73.2 (9.9) | 73.0 (10.1) |
| Sex (female) | 13 585 (36.6%) | 1897 (28.1%) | 11 688 (38.5%) |
| Smoking* | 4221 (12.7%) | 755 (12.3%) | 3466 (12.7%) |
| SBP (mm Hg) | 132.8 (16.1) | 130.6 (15.6) | 133.3 (16.2) |
| DBP (mm Hg) | 75.3 (10.7) | 73.9 (10.7) | 75.6 (10.7) |
| Total cholesterol (mmol/L)* | 4.5 (1.2) | 3.4 (0.7) | 4.8 (1.1) |
| LDL-C (mmol/L) | 2.7 (1.0) | 1.5 (0.3) | 3.0 (0.9) |
| Triglyceride (mmol/L)* | 1.6 (0.9) | 1.5 (1.0) | 1.6 (0.9) |
| Hypertension | 30 869 (83.2%) | 5649 (83.7%) | 25 220 (83.0%) |
| Diabetes | 12 544 (33.8%) | 3116 (46.2%) | 9428 (31.0%) |
| CHD | 37 120 (100.0%) | 6747 (100.0%) | 30 373 (100.0%) |
| AMI | 16 742 (45.1%) | 3740 (55.4%) | 13 002 (42.8%) |
| AMI past year | 1544 (4.2%) | 441 (6.5%) | 1103 (3.6%) |
| Stroke | 4452 (12.0%) | 921 (13.7%) | 3531 (11.6%) |
| Stroke past year | 1292 (3.5%) | 266 (3.9%) | 1026 (3.4%) |
| CVD | 19 183 (51.7%) | 4177 (61.9%) | 15 006 (49.4%) |
| Heart failure | 8888 (23.9%) | 1951 (28.9%) | 6937 (22.8%) |
| Atrial fibrillation | 8257 (22.2%) | 1807 (26.8%) | 6450 (21.2%) |
| Dementia | 1363 (3.7%) | 224 (3.3%) | 1139 (3.8%) |
| ASA | 23 431 (63.1%) | 4426 (65.6%) | 19 005 (62.6%) |
| Statin | 25 160 (67.8%) | 5735 (85.0%) | 19 425 (64.0%) |
| Simvastatin | 12 820 (34.5%) | 2810 (41.6%) | 10 010 (33.0%) |
| Pravastatin | 286 (0.8%) | 20 (0.3%) | 266 (0.9%) |
| Atorvastatin | 11 424 (30.8%) | 2723 (40.4%) | 8701 (28.6%) |
| Rosuvastatin | 998 (2.7%) | 235 (3.5%) | 763 (2.5%) |
| Other lipid-lowering drugs | 1231 (3.3%) | 227 (3.4%) | 1004 (3.3%) |
| Ezetimib | 953 (2.6%) | 175 (2.6%) | 778 (2.6%) |

Mean (SD) and frequencies (%).
*Missing data: smoking 3753, SBP 616, DBP 625, Total cholesterol 1510, Triglyceride 5366. For other variables there is no missing data.
AMI, acute myocardial infarction; ASA, acetylic salicylic acid; CHD, coronary heart disease; CVD, cardiovascular disease; DBP, diastolic blood pressure; LDL-C, low-density lipoprotein cholesterol; SBP, systolic blood pressure.

the patients with LDL-C >1.8, the corresponding estimated reductions were 22% and 18%. All-cause mortality for 5 years was estimated to decline by 6.4% (534/8344) when LDL-C was lowered to ≤1.8 mmol/L.

Sensitivity analyses by adding variables showed that the model was robust (see online supplemental table 1). When BMI and smoking status were included, the number of patients in the risk estimation cohort was reduced, from 52 042 to 9254. The proportion of patients who were predicted to suffer a CVD event was decreased from 25% to 22% but the relative reduction in number of events was the same. The results were similar without

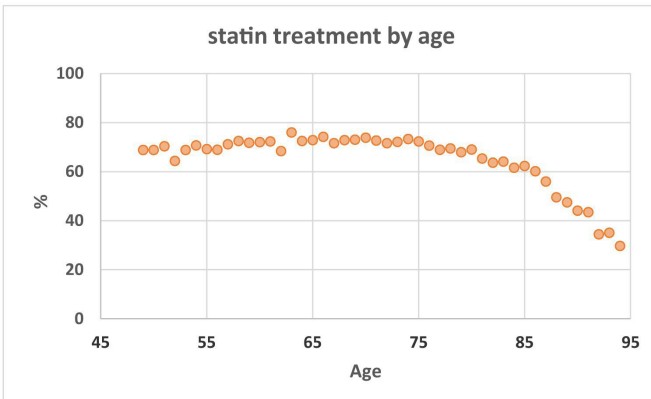

**Figure 1** Statin treatment by age. Included ages with more than 100 observations. The number of patients below the age of 50 is very few (603/37 120) as are the patients over the age of 90 (925/37 120).

including BMI and smoking status in the model. The same was true if the risk estimation model was based on the 16 577 patients with LDL-C (although LDL-C was not in the model). When systolic blood pressure was included in the model, the risk estimation cohort was decreased to 35 989 and the outcomes were relatively unchanged. Inclusion of statin treatment (0/1) in the model did not lead to any changes in the results.

The proportion of patients predicted to experience a CVD event was 24.6% (2937/11 960) among non-statin users and 24.9% (6271/25 160) among statin users, see figure 4. If all patients were prescribed 80 mg atorvastatin, the predicted proportion to experience a CVD event was reduced by 32.9% (967) among non-statin users and 5.4% (339) among statin users. The patients with statin treatment were younger (72.3 vs 74.5) and consisted of more men (68% vs 54%), had a diabetes diagnosis to a larger extent (36% vs 29%) and more often had a stroke or AMI during the past year (4%–5% vs 2.5%) (see online supplemental table 2).

Other lipid-lowering drugs were rarely used, 953/37 120 (2.6%) only 442 of whom were not also on statin. Patients

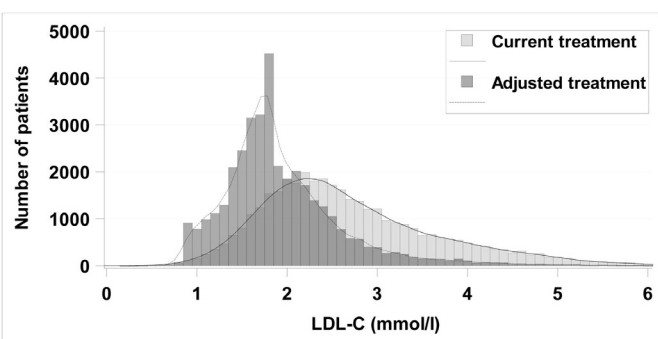

**Figure 2** LDL-C distribution in all patients in the study cohort with current treatment and adjusted treatment defined as if all patients received atorvastatin 80 mg (if less intense treatment before and LDL-C >1.8 mmol/L). LDL-C, low-density lipoprotein cholesterol.

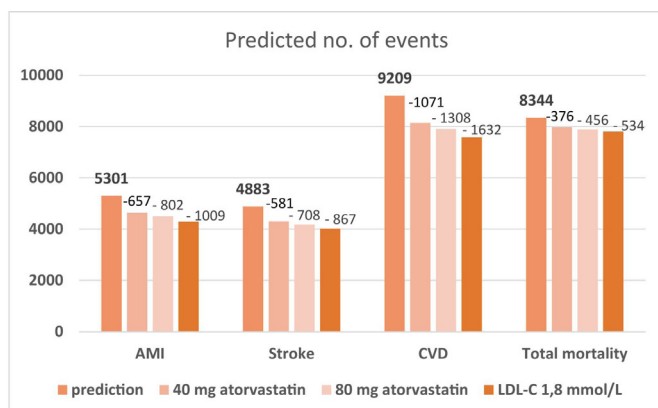

**Figure 3** Estimated number of events in the study cohort (37 120 patients) with registered levels of LDL-C and current treatment as well as when modelling a reduction in LDL-C to 1.8 mmol/L for all patients with higher LDL-C or an intensified statin lowering treatment with 80 mg atorvastatin for those with a less efficient treatment. AMI, acute myocardial infarction; CVD, cardiovascular disease; LDL-C, low-density lipoprotein cholesterol.

receiving other lipid-lowering drugs had slightly higher LDL-C than the rest of the study population.

## DISCUSSION
### Summary
One-third of our study population had no statin treatment and fewer than 1/5 reached the recommended LDL-C target of 1.8 mmol/L. Using data on the effect of statins from randomised trials, the number of CVD events was estimated to be substantially reduced if patients with no/low-dose statin were given atorvastatin 80 mg and even more so if all patients reached the LDL-C target. The largest gain seemed to result from initiating treatment in patients who are not receiving statin.

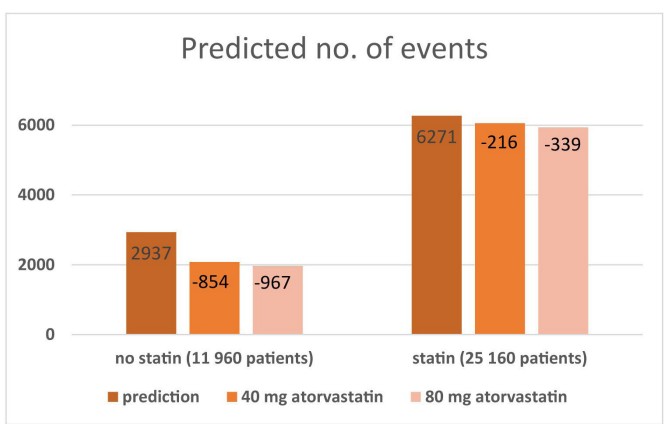

**Figure 4** Estimated number of events in the study cohort with registered levels of LDL-C and current treatment separated in patients with and without current statin treatment and the predicted number of events when applying a more intense statin treatment to all patients with a less efficient treatment. LDL-C, low-density lipoprotein cholesterol.

Ödesjö H, et al. BMJ Open 2020;**10**:e036920. doi:10.1136/bmjopen-2020-036920

## Strengths and limitations

The main strength of our study was the large number of primary care patients with CHD. Previous studies were often based on a post-MI population from secondary care registers. Another strength is the completeness of data concerning comorbidities and filled prescriptions. We based our risk model calculations on the same regional population as the predictions.

Information about LDL-C level was missing for many patients in the study cohort. These patients were older, diagnosed more often with atrial fibrillation and heart failure, less with diabetes and not as likely to be receiving statin treatment. Inclusion of them would probably have increased the predicted number of saved events. BMI, smoking and blood pressure were not included in the model due to the quantity of missing values in the risk estimation cohort. Sensitivity analyses showed that the results remained approximately the same without these variables.

It has been shown that compliance per se (placebo or active treatment) is associated with outcomes.[20] We assumed that all patients benefit proportionally from better adherence and improved treatment. A limitation of our study is therefore that included patients with suboptimal statin treatment might not respond to statin therapy in the same way as participants in the randomised trials in the meta-analysis used for risk reduction calculations. Furthermore, patients with no statin treatment could either have no prescription by a physician or be non-compliant. We were not able to adjust for this difference but since our aim was not to estimate causal effects but to get an estimation of the effects of a changed treatment pattern, we consider our results still valid.

We did not take competing risks into account and assumed that no patient had died from other causes during the follow-up period. Patients in our risk estimation cohort who died before a potential event were right censored and 9.2% died from other causes without a previous event. Our model appears to slightly overestimate the number of CVD events, but the percentage decrease is not necessarily affected.

## Comparison with existing literature

A recent Swedish study described short-term adherence to statins and potential effects of improved treatment among 5904 post-MI patients.[21] The study found an estimated 805–2262 CVD events in over 10 years among those who did not reach LDL-C target and predicted that a reduction of LDL-C to 1.8 mmol/L would lower the number by 132–343 (15%–16%). The patients in that study were younger, which might explain the smaller predicted reduction in events.

LDL-C is monitored 1 year after AMI in the Swedish national quality register Registry of Secondary Preventive Care after Cardiac Infarction (SEPHIA). In 2015, mean LDL-C was 1.99 mmol/L and 51% had LDL-C <1.8 mmol/L.[22] Lipid-lowering treatment after AMI is gradually increasing in Sweden.[15] A Norwegian study of 42 707 patients discharged from hospital after AMI in 2009–2013

reported that 84% were being treated with statins 1 year later.[8] Mean LDL-C was 2.7 mmol/L in our study cohort and only 18% had LDL-C ≤1.8 mmol/L also including those younger than 75 as those in SEPHIA. This difference between improved treatment in the short term and our finding of poor control in the long term may be due to poorer adherence to lipid-lowering treatment over time and the fact that new guidelines are applied to patients with a recent AMI event. Poor long-term adherence has been demonstrated earlier with discontinuation rates of 67% over 5 years.[23] The probability of persistence has increased over the years but remains low already a few years after initiation of treatment.[24] Since our study is based on an unselected regional primary care patient population, it seems plausible that the results are generalisable to regions and countries with a similar population.

Over the past few years, statin treatment has proven to be safe and effective among the aged as well, but the proportion of patients in our study receiving it diminishes after the age of 75.[2] The importance of continued statin treatment is further stressed by the fact that LDL-C remains an important risk factor for CVD even at high age as opposed to other traditional risk factors.[1]

Our cohort consists of primary care patients with a CHD diagnosis. It has been shown that patients in primary care with known CVD from diagnoses in hospital but without a registered one in primary care receive less statin treatment.[25] In light of these results, our study presumably underestimates the potential event reduction by means of improved treatment.

The reasons for non-adherence to guidelines may depend on the physician, patient or organisational factors. A recent Swedish study showed that patients with concomitant use of other cardioprotective medications or smoking had higher adherence to refill and lipid-lowering treatment.[26] Two Danish studies found adherence to statin treatment to be lower among patients with a more risk-seeking attitude and physicians who did not assess treatment after initiation.[27 28] The reasons that physicians do not adhere to lipid-lowering guidelines may be the same as for the blood pressure target: acceptance of higher values than recommended, competing medical problems.[29] Regardless of reasons for non-adherence, our study shows that there is room for improvement of secondary preventive lipid-lowering therapy.

## Implications for research and/or practice

In this unselected primary care population of patients with CHD in a large Swedish region, a high proportion of patients do not reach LDL-C target. Assuming the same benefits of intensified statin treatment as in the CTT meta-analyses, there is a significant potential to reduce the number of events if treatment guidelines were to be followed to a greater extent. The results of this cross-sectional study support further studies of the effect of increased compliance with recommendations.

**Contributors** HÖ, SB, SF, PH, KM, AR and SAE contributed to the conception and design of the work. All authors contributed to the acquisition, analysis, or

interpretation of data for the work. HÖ drafted the manuscript. All authors critically revised the manuscript. All authors gave final approval and agree to be accountable for all aspects of work ensuring integrity and accuracy.

**Funding** The authors disclosed receipt of the following financial support for the research, authorship, and/or publication of this article: This work was supported by Närhälsan R&D Health Care, R&D Centre Gothenburg and Södra Bohuslän for postgraduate studies. This work was also supported by grants from the Swedish government under the Agreement concerning Research and Education of Doctors (grant number ALFGBG-717211), the Swedish Heart and Lung Foundation (grant number 2018-0366) and the Swedish Research Council (grant number 2018-02527).

**Competing interests** SAE is employed by AstraZeneca since September 2019 but was affiliated to University of Gothenburg during the main part of the study.

**Patient and public involvement** Patients and/or the public were not involved in the design, or conduct, or reporting, or dissemination plans of this research.

**Patient consent for publication** Not required.

**Provenance and peer review** Not commissioned; externally peer reviewed.

**Data availability statement** No data are available. No additional unpublished data are available.

**ORCID iD**
Helena Ödesjö http://orcid.org/0000-0002-0997-3699

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
