## [Reviewer comments · BMJ Open]

ARTICLE DETAILS

TITLE (PROVISIONAL)	Adherence to lipid-lowering guidelines for secondary prevention and potential reduction in CVD events in Swedish primary care – a cross-sectional study
AUTHORS	Ödesjö, Helena; Björck, Staffan; Franzén, Stefan; Hjerpe, Per; Manhem, Karin; Rosengren, Annika; Thorn, Jörgen; Adamsson Eryd, Samuel

VERSION 1 – REVIEW

REVIEWER	Engi Algharably Charite-Universitaetsmedizin
REVIEW RETURNED	18-Feb-2020

GENERAL COMMENTS	Review Comments to the Author I would appreciate getting an opportunity to review this important paper. In this work, the authors performed a cross-sectional study in a large cohort of patients in primary care registered in a national registry with a history of previous coronary heart disease and with indication to receive lipid lowering therapy for secondary prevention. They assessed the proportion of patients according to available patients' data who had their LDL-C level controlled up to the values recommended by the most recent dyslipidemia guidelines. The authors also synthesized and validated a risk predictive model for development of CVD events based on large subset of patients derived from the same registry. They found that almost two thirds of patients were treated with statin drugs but only less than a fifth of the total study cohort have controlled serum levels of LDL-C (≤ 1.8 mmol/L). They estimated a 5-year risk of CVD to be 9,209/37,120 and predicted a 14% risk reduction by more intensive statin therapy. The predicted risk reduction was higher (18%) when adjusted the lower LDL-C target. A substantial risk reduction of CVD events could be obtained in patients who did not receive statins in primary care than in patients who required statin intensification. This is an important topic and the present paper will contribute future works in addressing the problem of suboptimal use of lipid lowering therapy especially in secondary prevention of CVD and the potential of CVD protection that could be attained by adherence to target levels. However, this paper still has some points which need further consideration. Please see my comments for revision work. Introduction The introduction is nicely and concisely written but falls a bit short in comprehensively covering the topics especially those relevant to statin treatment e.g. reported benefits of risk reduction in both mortality and CVD events based on meta-analyses, the risk-benefit ratio especially in problematic patient populations (e.g. elderly), the wide spread phenomenon of suboptimal use of statins in general
--

	practice and the perceived barriers to statin therapy in long term since this provides also a rationale for the study and later as implications for further research. It would be more informative if more of literature review was added. Methods  - Using the last observation carried forward method could be added to the limitation of the study since the authors used this method to impute missing data for long follow up periods (up to 900 days) and the extent of missing values is not given. The last observation carried forward method is not reliable for statistical robustness when accounting for missing data without the risk of bias. - The non-inclusion of statin treatment in the risk prediction model is for me unclear since it is one predictive variable used in the predicted risk assessment performed for the individuals of the study cohort. Was LDL-C level included in the model? Results  - The authors identified in the subset of patients who developed CV event in the last year, that a major fraction of them (78%) were also treated with statin but only 25% reaching the target LDL-C level. It would be very helpful if the pattern of statin therapy showing the inappropriateness of dose/type with regard to risk profile was presented. - In table 1: Is there a specific reason why pravastatin was the least prescribed statin in the cohort? Quantitative variables (BP, LDL-C,...etc) should be indicated in the footnote as: expressed as mean (SD) Please be consistent in the way numbers are presented; using a comma separator in figures of thousands or not in the whole manuscript - Please include a table to summarize the performed sensitivity analysis in a supplementary file if possible. Discussion.  - Please discuss and stress the importance of adherence to guidelines in treating dyslipidemia in the elderly patient since the prevalence of atherosclerosis increases with age and the number of cardiovascular events rises in older patients noting that the average age in this cohort is 73 years. - The point of long term adherence should also be stressed because the advantages/benefits of statin treatment become evident after at least one year. Please provide relevant studies to compare the long term adherence to statins in similar/comparable population if possible. - Please refer to and discuss the interesting finding of the nearly equal ratio of patients predicted to experience a CVD event among non-statin users versus statin users (24.6 % vs 24.9 %). This implies profoundly that, based on model predictions, statin therapy is at most underused in the long term, could the reasons be drug interactions potential, statin intolerance...etc? An appropriateness pattern of statin therapy would be mainly helpful understanding the gap, perhaps as a starting point for further qualitative and quantitative research research which is suggested as implication for future studies.
--	---

REVIEWER	John Abramson Harvard Medical School Boston, MA, USA
REVIEW RETURNED	08-Apr-2020

GENERAL COMMENTS

2. The abstract should state that the calculated reduction in CVD events is based upon applying rate reductions noted in the CTT meta-analyses of RCTs to the non-adherent population.

3. The study design only answers the research question--potential reduction in number of rate and number of events with improved treatment and lowered LDL-C--if the rate reductions noted in the CTT meta-analyses of statin treatment vs control groups are the same or similar to the difference between the natural population of people not meeting treatment or LDL goals and those who are. As discussed below there is strong evidence that CTT control groups from RCTs and the non-adherent natural population are not the same.

6. The outcomes--potential number and percentage of events that could be prevented by compliance with guidelines of the non-complying population--is based upon the assumption that the adherent and non-adherent populations are similar. No evidence is presented to support this assumption and strong evidence exists to rebut it. See 8 below

8. References are up to date, except regarding the issue discussed above. The following article, NEJM 1980, shows that cholesterol-lowering with clofibrate did not reduce mortality compared to placebo, but patients who were at least 80% compliant with clofibrate had significantly lower five year mortality rate than those who were not: 15.0 vs. 24.6 percent. The important finding with regard to the study under review showed that people who complied with their placebo therapy had far lower five year mortality rate than those who did not: 15.1 vs 28.3 percent ($p=4.7 \times 10^{-16}$). This provides overwhelming evidence that people who comply with therapy--even placebo therapy--experience significantly lower event rates, and proves that a natural experiment based on compliance vs. non-compliance creates two different and non-comparable populations. The submitted study is based on the assumption that the compliant and non-compliant populations are fundamentally the same, and therefore simply adding appropriate statin therapy to the non-compliant population would provide the benefit reported by CTT. Before the conclusions in this article can be accepted, this assumption must be substantiated, and results from the NEJM article addressed.

[Group CDPR. Influence of adherence to treatment and response of cholesterol on mortality in the Coronary Drug Project. New England Journal of Medicine. 1980;303(18):1038-1041.]

9. The results section fails to inform readers that reported benefits of adherence to treatment guidelines are valid only if the benefit of adding statin therapy to the naturally selected population that is not in compliance with the guidelines is substantially the same as the benefit of adding statin therapy to the randomly assigned control groups in the RCTs included in the CTT meta-analyses.

10, 11, 12: Discussion, conclusion, and limitations are incomplete and misleading without a full discussion of the above assumption and the evidence against its validity.

14. If SA Eryd was paid as an employee of AstraZeneca for the time spent writing this article, this should be declared in the funding of the article and perhaps in the institutional affiliations noted in the Complete List of Authors (per BMJ editorial policy)

VERSION 1 – AUTHOR RESPONSE

Reviewer: 1 Reviewer Name: Engi Algharably Institution and Country: Charite-Universitaetsmedizin
Please state any competing interests or state 'None declared': none Please leave your comments for the authors below Review Comments to the Author I would appreciate getting an opportunity to review this important paper. In this work, the authors performed a cross-sectional study in a large cohort of patients in primary care registered in a national registry with a history of previous coronary heart disease and with indication to receive lipid lowering therapy for secondary prevention. They assessed the proportion of patients according to available patients' data who had their LDL-C level controlled up to the values recommended by the most recent dyslipidemia guidelines. The authors also synthesized and validated a risk predictive model for development of CVD events based on large subset of patients derived from the same registry. They found that almost two thirds of patients were treated with statin drugs but only less than a fifth of the total study cohort have controlled serum levels of LDL-C (≤ 1.8 mmol/L). They estimated a 5-year risk of CVD to be 9,209/37,120 and predicted a 14% risk reduction by more intensive statin therapy. The predicted risk reduction was higher (18%) when adjusted the lower LDL-C target. A substantial risk reduction of CVD events could be obtained in patients who did not receive statins in primary care than in patients who required statin intensification. This is an important topic and the present paper will contribute future works in addressing the problem of suboptimal use of lipid lowering therapy especially in secondary prevention of CVD and the potential of CVD protection that could be attained by adherence to target levels. However, this paper still has some points which need further consideration. Please see my comments for revision work.

Introduction The introduction is nicely and concisely written but falls a bit short in comprehensively covering the topics especially those relevant to statin treatment e.g. reported benefits of risk reduction in both mortality and CVD events based on meta-analyses, the risk-benefit ratio especially in problematic patient populations (e.g. elderly), the wide spread phenomenon of suboptimal use of statins in general practice and the perceived barriers to statin therapy in long term since this provides also a rationale for the study and later as implications for further research. It would be more informative if more of literature review was added. - The introduction has been supplemented accordingly

Methods - Using the last observation carried forward method could be added to the limitation of the study since the authors used this method to impute missing data for long follow up periods (up to 900 days) and the extent of missing values is not given. The last observation carried forward method is not reliable for statistical robustness when accounting for missing data without the risk of bias. - We acknowledge the limitations with LOCF and we have extended the description in the methods section. Most values are captured at yearly visits and we consider the alternatives less suitable with imputation or accepting more missing patients. The risks with imputation are that intra-individual correlation is not taken into account. In the models only LDL-C is used of the LOCF collected medical variables and the register (QregPV) reports these data also 900 days back in time. - The non-inclusion of statin treatment in the risk prediction model is for me unclear since it is one predictive variable used in the predicted risk assessment performed for the individuals of the study cohort. Was LDL-C level included in the model? - No, LDL-C was not included in the model since we cannot effectively estimate the effect of LDL-C (or statin) due to lack of information on other confounders in these observational data. LDL-C consistently shows not a linear, but a J-shaped relationship with the cardiovascular endpoints in observational studies since low cholesterol is linked to other risk factors such as malnutrition and not yet diagnosed disease. We can't estimate the causal effect of LDL-C lowering or statin treatment but an association between LDL-C and risk. We have instead chosen to estimate the general risk level based on primarily age and sex and studied the estimated changes in the risk level when changing LDL-C/giving statin giving us an estimate of the effect and the risk level when treatment is altered.

Results - The authors identified in the subset of patients who developed CV event in the last year, that a major fraction of them (78%) were also treated with statin but only 25% reaching the target LDL-C level. It would be very helpful if the pattern of statin therapy showing the inappropriateness of dose/type with regard to risk profile was presented.

- We regard the studied population as quite homogenous since previous ischemic heart disease puts all patients in a very high-risk group according to current guidelines. We have though presented data on statin use in relation to age. Since we are studying a primary care population some patients had a recent event and some had an event many years ago. We have not analysed this in any detail but as presented, the group with a recent event were on statin treatment to a larger extent than the whole study population, although still less than 80%. - In table 1: Is there a specific reason why pravastatin was the least prescribed statin in the cohort? - Yes, Swedish national guidelines recommended simvastatin until 2015 and thereafter atorvastatin (the patent for Lipitor/atorvastatin expired in 2012 in Sweden). Rosuvastatin or pravastatin has never been a first or second choice and thereby used to a very small extent. Quantitative variables (BP, LDL-C,...etc) should be indicated in the footnote as: expressed as mean (SD) - Agreed, done accordingly Please be consistent in the way numbers are presented; using a comma separator in figures of thousands or not in the whole manuscript - Agreed, done accordingly - Please include a table to summarize the performed sensitivity analysis in a supplementary file if possible. - Included table as supplementary material Discussion. - Please discuss and stress the importance of adherence to guidelines in treating dyslipidemia in the elderly patient since the prevalence of atherosclerosis increases with age and the number of cardiovascular events rises in older patients noting that the average age in this cohort is 73 years. - We agree that this is important to stress. References are now added concerning this topic. - The point of long term adherence should also be stressed because the advantages/benefits of statin treatment become evident after at least one year. Please provide relevant studies to compare the long term adherence to statins in similar/comparable population if possible. - We agree that this is important to stress. References are now added concerning this topic. - Please refer to and discuss the interesting finding of the nearly equal ratio of patients predicted to experience a CVD event among non-statin users versus statin users (24.6 % vs 24.9 %). This implies profoundly that, based on model predictions, statin therapy is at most underused in the long term, could the reasons be drug interactions potential, statin intolerance...etc? An appropriateness pattern of statin therapy would be mainly helpful understanding the gap, perhaps as a starting point for further qualitative and quantitative research which is suggested as implication for future studies. - It is likely that patients with statin treatment had more apparent reasons for treatment and to remain on treatment, such as diabetes, higher burden of previous disease and salicylic acid treatment as shown in supplement.

Reviewer: 2 Reviewer Name: John Abramson Institution and Country: Harvard Medical School Boston, MA, USA Please state any competing interests or state 'None declared': I have served as a plaintiffs' expert in litigation regarding Crestor Please leave your comments for the authors below 2. The abstract should state that the calculated reduction in CVD events is based upon applying rate reductions noted in the CTT meta-analyses of RCTs to the non-adherent population. - Done accordingly 3. The study design only answers the research question--potential reduction in number of rate and number of events with improved treatment and lowered LDL-C--if the rate reductions noted in the CTT meta-analyses of statin treatment vs control groups are the same or similar to the difference between the natural population of people not meeting treatment or LDL goals and those who are. As discussed below there is strong evidence that CTT control groups from RCTs and the non-adherent natural population are not the same. - We agree that this is plausible, a known problem with RCTs. 6. The outcomes--potential number and percentage of events that could be prevented by compliance with guidelines of the non-complying population--is based upon the assumption that the adherent and non-adherent populations are similar. No evidence is presented to support this assumption and strong evidence exists to rebut it. See 8 below - The fact that the patient has not filled a prescription is not necessary a consequence of the patient being non-compliant, it might instead be a consequence of the physician not having prescribed the drug at all or a lower dose than needed by active choice or inertia. We do not consider it likely that the majority of these patients are non-compliant but of course we can't know for sure. 8. References are up to date, except regarding the issue discussed above.

The following article, NEJM 1980, shows that cholesterol-lowering with clofibrate did not reduce mortality compared to placebo, but patients who were at least 80% compliant with clofibrate had significantly lower five year mortality rate than those who were not: 15.0 vs. 24.6 percent. The important finding with regard to the study under review showed that people who complied with their placebo therapy had far lower five year mortality rate than those who did not: 15.1 vs 28.3 percent ($p=4.7 \times 10^{-16}$). This provides overwhelming evidence that people who comply with therapy-- even placebo therapy-- experience significantly lower event rates, and proves that a natural experiment based on compliance vs. non-compliance creates two different and non-comparable populations. The submitted study is based on the assumption that the compliant and non-compliant populations are fundamentally the same, and therefore simply adding appropriate statin therapy to the noncompliant population would provide the benefit reported by CTT. Before the conclusions in this article can be accepted, this assumption must be substantiated, and results from the NEJM article addressed. [Group CDPR. Influence of adherence to treatment and response of cholesterol on mortality in the Coronary Drug Project. New England Journal of Medicine. 1980;303(18):1038-1041.] - Firstly, we want to point out the fact that our study is not based on compliant vs noncompliant patients. In the group where treatment is adjusted, patients that have either not filled a prescription or filled a prescription (of a statin and dose) less potent than atorvastatin 80 mg and have an LDL-C \geq 1.8 mmol/L, are included. Patients that are non-compliant can be so for many different reasons that we cannot control for. We have therefor included the suggested reference and pointed out the problem in the discussion. We find it reasonable to believe that the order of magnitude of improvement with more effective treatment is reasonably accurate. As discussed above, the fact that a patient has not filled a prescription is not the same as being non-compliant. 9. The results section fails to inform readers that reported benefits of adherence to treatment guidelines are valid only if the benefit of adding statin therapy to the naturally selected population that is not in compliance with the guidelines is substantially the same as the benefit of adding statin therapy to the randomly assigned control groups in the RCTs included in the CTT meta-analyses. - We have chosen to discuss this in the discussion section and not in the results section. 10, 11, 12: Discussion, conclusion, and limitations are incomplete and misleading without a full discussion of the above assumption and the evidence against its validity. - We hope that you'll find that the corrections according to comments above have reasonably overcome this problem. 14. If SA Eryd was paid as an employee of AstraZeneca for the time spent writing this article, this should be declared in the funding of the article and perhaps in the institutional affiliations noted in the Complete List of Authors (per BMJ editorial policy) - S Adamsson Eryd was not employed by AstraZeneca during the work with this article. He is now employed by AstraZeneca (since September 2019) but was affiliated to Gothenburg University during the work with this study

VERSION 2 – REVIEW

REVIEWER	Engi Algharably Charité – Universitätsmedizin Berlin
REVIEW RETURNED	24-Jul-2020
GENERAL COMMENTS	to improve the quality and layout of the figures especially data on the y-axis. numbers should be written out at the beginning of a sentence e.g. page 14 line 289
REVIEWER	John Abramson Harvard Medical School, Boston, MA USA I have served as a plaintiffs' expert in litigation involving statins, but

	have not done so since 2017. I am writing a book about the quality of information available to physicians.
REVIEW RETURNED	09-Jul-2020

GENERAL COMMENTS	This paper is, as described in the title, two studies combined: First “an observational cross-sectional register study of all primary care patients with CHD in a large Swedish region” that determined the “Proportion of patients on statin treatment and proportion of patients achieving LDL-C ≤ 1.8 mmol/L.” Second, based on the proportion of undertreated patients and the estimated risk in the population, the study “shows that the number of CVD events could be substantially reduced if patients with no/low-dose statin were given atorvastatin 80 mg and even more so if all patients reached the LDL-C target.” The results of the first part of the study are of interest. The second part of the study is based on an unproven syllogism: A. Many patients in Swedish general practices with CHD are not reaching their LDL target B. CTT meta-analyses show that secondary prevention patients treated with statins in RCTs have significant reduction in CVD events compared to those in control groups. Therefore: C. Treating Swedish patients with CHD not reaching their LDL target could substantially reduce the number of CVD events. This syllogism cannot simply be posited as true. First, many of the patients in the RCTs included in the CTT meta-analyses, including HPS and JUPITER (the two largest trials), were screened during a run-in period precisely for compliance with prescribed therapy. Second, no study that I am aware of has tested the effect of statin therapy in a non-complying population to see if the benefit is similar to that documented in the CTT meta-analyses. Third, as you have mentioned briefly and cited in Footnote 20, non-compliers with cholesterol-lowering therapy—whether assigned to active therapy or placebo—have been documented to have much worse outcomes than compliers. There is not evidence that the population of participants screened and accepted into the studies included in the CTT meta-analyses are similar to and would react to statin therapy the same way as those who are not meeting their targets in the general population. Nonetheless, the submitted article makes numerous statements that assume such evidence of benefit in non-compliers has been established:  • Abstract  o Objectives: “The aim was to...predict potential reductions...” o Primary and secondary outcome measures: “Predicted number of CVD events calculated for...b) improved treatment and c) lowered LDL-C, based on applying rate reductions from a meta-analysis of RCTs to the potentially undertreated population. o Results: “If all patients without a statin or with less potent statin treatment were given atorvastatin 80 mg, a reduction of CVD events by 14% (7,901 vs 9,209) was predicted. If all patients achieved LDL-C ≤ 1.8 mmol/L the predicted number of events was reduced by 18% (7,577 vs 9,209).” o Conclusion: “By improving adherence to treatment guidelines, a substantial reduction in CVD events among patients with established CHD in primary care could be achieved. o Strengths and limitations of this study: No mention is made of the absence of evidence that treating non-complying patients with statins will produce results that are similar to those achieved with the
---

	context of the RCTs included in the CTT meta-analysis.  • Full manuscript  o Results: “Total predicted CVD events for 5 years in the study cohort was 9,209 (24.8%), see Figure 3. A decrease of LDL-C to the target (≤ 1.8 mmol/l) or intensified statin treatment (all patients with less efficient therapy receive atorvastatin 80 mg) would result in a reduction of expected CVD events for 5 years by 1,632 and 1,308. This would result in a decreased event risk of 18% and 14%. Including only the patients with LDL-C > 1.8, the corresponding reductions were 22% and 18%. All-cause mortality for 5 years declined by 6.4 % (534/8,344) when LDL-C was lowered to ≤ 1.8 mmol/L.” o Discussion: “Summary: This study shows that the number of CVD events could be substantially reduced if patients with no/low-dose statin were given atorvastatin 80 mg and even more so if all patients reached the LDL-C target.” o Strengths and limitations: “The patients with appropriate treatment might differ from those without as can also be a problem in RCTs. It has been shown that compliance per se (placebo or active treatment) is associated with lower mortality [20].” This comment acknowledges a recommendation I made in the first review, but gives this only cursory mention, not the suggestion that the population of non-compliant patients (or doctors) might not benefit from statin therapy in the same way as participants in controlled trials of statins. o Comparison with existing literature: “Our results suggest a large potential for improvement of secondary prevention in primary care by improved statin treatment and especially initiate treatment in non-statin users.” This statement is based on the unproven syllogism and is not supported by the evidence presented. It could be restated to say that the high proportion of people with CHD who are not reaching LDL-C target warrants further study to determine if meeting the recommended target would provide benefits consistent with the CTT meta-analyses. o Implications for research and/or practice: “The treatment gap for lipid-lowering therapy in a primary care population with established CHD was large, representing a significant potential to reduce the number of events if treatment guidelines were to be followed to a greater extent.” For all the reasons stated above, this statement is not supported by evidence. The results of the cross-sectional study show a high prevalence of patients with CHD not meeting recommended LDL-C target, which supports further study of the effect of additional statin therapy to bring such patients into compliance with recommendations would be beneficial.
--	--

VERSION 2 – AUTHOR RESPONSE

Reviewer1

Reviewer Name: Engi Algharably

Institution and Country: Charité – Universitätsmedizin Berlin

Reviewer 1 comments: to improve the quality and layout of the figures especially data on the y-axis. numbers should be written out at the beginning of a sentence e.g. page 14 line 289

- **Our response:** *We have made changes accordingly*

Reviewer 2

Reviewer Name: John Abramson

Institution and Country: Harvard Medical School, Boston, MA USA

- **Overall response:** *We have further balanced conclusions vs actual findings. We have rewritten parts of the manuscript in order to specifically address limitations of the study design.*

Reviewer 2 comments: This paper is, as described in the title, two studies combined: First “an observational cross-sectional register study of all primary care patients with CHD in a large Swedish region” that determined the “Proportion of patients on statin treatment and proportion of patients achieving LDL-C ≤ 1.8 mmol/L.” Second, based on the proportion of undertreated patients and the estimated risk in the population, the study “shows that the number of CVD events could be substantially reduced if patients with no/low-dose statin were given atorvastatin 80 mg and even more so if all patients reached the LDL-C target.”

The results of the first part of the study are of interest. The second part of the study is based on an unproven syllogism:

- A. Many patients in Swedish general practices with CHD are not reaching their LDL target
- B. CTT meta-analyses show that secondary prevention patients treated with statins in RCTs have significant reduction in CVD events compared to those in control groups.

Therefore:

- C. Treating Swedish patients with CHD not reaching their LDL target could substantially reduce the number of CVD events.

This syllogism cannot simply be posited as true. First, many of the patients in the RCTs included in the CTT meta-analyses, including HPS and JUPITER (the two largest trials), were screened during a run-in period precisely for compliance with prescribed therapy. Second, no study that I am aware of has tested the effect of statin therapy in a non-complying population to see if the benefit is similar to that documented in the CTT meta-analyses. Third, as you have mentioned briefly and cited in Footnote 20, non-compliers with cholesterol-lowering therapy—whether assigned to active therapy or placebo—have been documented to have much worse outcomes than compliers.

There is not evidence that the population of participants screened and accepted into the studies included in the CTT meta-analyses are similar to and would react to statin therapy the same way as those who are not meeting their targets in the general population.

Nonetheless, the submitted article makes numerous statements that assume such evidence of benefit in non-compliers has been established:

- Abstract

o Objectives: “The aim was to...predict potential reductions...”

Our response: *We have changed to “...provide an estimate...”*

o Primary and secondary outcome measures: “Predicted number of CVD events calculated for...b) improved treatment and c) lowered LDL-C, based on applying rate reductions from a meta-analysis of RCTs to the potentially undertreated population.

- **Our response:** *Predicted changed to estimated*

o Results: “If all patients without a statin or with less potent statin treatment were given atorvastatin 80 mg, a reduction of CVD events by 14% (7,901 vs 9,209) was predicted. If all patients achieved LDL-C ≤ 1.8 mmol/L the predicted number of events was reduced by 18% (7,577 vs 9,209).”

- **Our response:** *Also here predicted removed and replaced by estimated*

o Conclusion: “By improving adherence to treatment guidelines, a substantial reduction in CVD events among patients with established CHD in primary care could be achieved.

- **Our response:** *Changed to: “Based on the assumption that included patients would react to statin therapy the same way as the patients in randomized trials, improved adherence to treatment guidelines could lead to a substantial reduction in new CVD events.”*

o Strengths and limitations of this study: No mention is made of the absence of evidence that treating non-complying patients with statins will produce results that are similar to those achieved with the context of the RCTs included in the CTT meta-analysis.

- **Our response:** *A bullet point is added: “Whether patients in our study react to statin therapy similarly as in the randomized trials in the meta-analysis used is not known.”*

- Full manuscript

o Results: “Total predicted CVD events for 5 years in the study cohort was 9,209 (24.8%), see Figure 3. A decrease of LDL-C to the target (≤ 1.8 mmol/l) or intensified statin treatment (all patients with less efficient therapy receive atorvastatin 80 mg) would result in a reduction of expected CVD events for 5 years by 1,632 and 1,308. This would result in a decreased event risk of 18% and 14%. Including only the patients with LDL-C > 1.8 , the corresponding reductions were 22% and 18%. All-cause mortality for 5 years declined by 6.4 % (534/8,344) when LDL-C was lowered to ≤ 1.8 mmol/L.”

- **Our response:** *Also, here we emphasize that our calculations are estimations and hopefully in the light of all other changes performed it is clear to reader under what assumptions these results should be interpreted.*

o Discussion: “Summary: This study shows that the number of CVD events could be substantially reduced if patients with no/low-dose statin were given atorvastatin 80 mg and even more so if all patients reached the LDL-C target.”

- **Our response:** *We have emphasized the descriptive results and placed it first in the initial section of the discussion. After all, our study is a description of findings in a 1.6 million population. Thereafter wordings are changed:*

“Using data on the effect of statins from randomized trials, the number of CVD events was estimated to be substantially reduced if patients with no/low-dose statin were given atorvastatin 80 mg and even more so if all patients reached the LDL-C target.”

Limitations of this approach is further commented later in the discussion.

o Strengths and limitations: “The patients with appropriate treatment might differ from those without as can also be a problem in RCTs. It has been shown that compliance per se (placebo or active treatment) is associated with lower mortality [20].” This comment acknowledges a recommendation I made in the first review, but gives this only cursory mention, not the suggestion that the population of non-compliant patients (or doctors) might not benefit from statin therapy in the same way as participants in controlled trials of statins.

- **Our response:** *We have rewritten and reorganized this paragraph and most importantly added:*

“A limitation of our study is therefore that included patients with suboptimal statin treatment might not respond to statin therapy in the same way as participants in the randomized trials in the meta-analysis used for risk reduction calculations.”

o Comparison with existing literature: “Our results suggest a large potential for improvement of secondary prevention in primary care by improved statin treatment and especially initiate treatment in non-statin users.” This statement is based on the unproven syllogism and is not supported by the evidence presented. It could be restated to say that the high proportion of people with CHD who are not reaching LDL-C target warrants further study to determine if meeting the recommended target would provide benefits consistent with the CTT meta-analyses.

- **Our response:** *The last sentence in this section mentioned above has been removed and exchanged for: “Regardless of reasons for non-adherence, our study shows that there is room for improvement of secondary preventive lipid lowering therapy.”*

In relation to the very poor guidelines adherence, this is now a mere reflection around the descriptive results.

o Implications for research and/or practice: “The treatment gap for lipid-lowering therapy in a primary care population with established CHD was large, representing a significant potential to reduce the number of events if treatment guidelines were to be followed to a greater extent.” For all the reasons stated above, this statement is not supported by evidence. The results of the cross-sectional study show a high prevalence of patients with CHD not meeting recommended LDL-C target, which supports further study of the effect of additional statin therapy to bring such patients into compliance with recommendations would be beneficial.

- **Our response:** *We have exchanged the sentence to a slightly modified version of the suggested:*

“Assuming the same benefits of intensified statin treatment as in the CTT meta-analyses, there is a significant potential to reduce the number of events if treatment guidelines were to be followed to a greater extent. The results of this cross-sectional study support further studies of the effect of increased compliance with recommendations.”